# Effects of Invasive Smooth Cordgrass Degradation on Avian Species Diversity in the Dafeng Milu National Nature Reserve, a Ramsar Wetland on the Eastern Coast of China

**Taiyu Chen** [1], **Pan Chen** [2], **Bing Liu** [3], **Dawei Wu** [1] and **Changhu Lu** [1,*]

1   College of Life Sciences, Nanjing Forestry University, Nanjing 210037, China; chenty@njfu.edu.cn (T.C.)
2   College of Life Sciences, Anhui Normal University, Wuhu 241000, China
3   Dafeng Milu National Nature Reserve, Yancheng 224100, China
*   Correspondence: luchanghu@njfu.com.cn

**Abstract:** Invasive smooth cordgrass (*Spartina alterniflora*) has been expanding rapidly through the coastal wetlands of eastern China and these changes negatively affect local birds. In the Dafeng Milu National Nature Reserve (henceforth referred to as DMNNR), rapid degradation of spartina occurs after an increase in milu (*Elaphures davidianus*; hereafter elk) numbers and ecological hydrological engineering. We evaluated the impact of such degradation on the abundance and species diversity of birds in the DMNNR during 2017–2021. We found that the area covered by *S. alterniflora* decreased significantly in the study area at a rate of 310 ha per year and by 62% during 2017–2021 ($p < 0.01$). With this decrease in the *S. alterniflora* area, the species richness and abundance of birds first increased and then decreased. Songbird density clearly decreased but species richness did not significantly do so. This research demonstrated that during the initial stages of vegetation degradation, there was a positive effect on bird diversity. With the increasing vegetation degradation increases, both songbirds and waterbirds experience negative impacts. The DMNNR is an important stopover site for waterbirds in the East Asian–Australasian Flyway, and additional measures are needed to control vegetation degradation and to restore the native habitats for birds.

**Keywords:** elk; smooth cordgrass; vegetation degradation; bird diversity; coastal wetlands

## 1. Introduction

The invasion of exotic plants often changes local vegetation communities and affects the number and diversity of bird populations by changing the structure of food webs [1–3]. Many studies have shown that invasion by *Spartina alterniflora* (hereafter spartina) has resulted in severe declines in bird species numbers and abundance in local habitats [2,4]. In the Yellow River Delta, the number of bird species in the nonspartina area was greater than that in the spartina area, and the population density of birds in the spartina community were significantly lower than those in other habitat types [4]. Habitat loss and deterioration are the main reasons for the decline in bird diversity [5]. Although certain songbirds and breeding birds use and even prefer spartina-invaded habitats [6], their densities are lower than those in the native *Phragmites australis* habitats [4].

Due to the negative impact of the spartina invasion on local ecosystems, different measures have been taken worldwide to eliminate this plant from where it is invading [7,8]. The number of shorebirds, geese, and ducks in the Shanghai Chongming Dongtan wetland in China has been effectively restored through manual removal and waterlogging of spartina [9]. However, between 2005 and 2011, during the period of invasive spartina eradication, there was a significant decline of nearly 50% in populations of the federally endangered California clapper rail (*Rallus longirostris obsoletus*) in San Francisco Bay [10]. Rapid action for eradication of the invasive spartina plant is crucial for its population reduction, prevention of further dissemination, and, ultimately, complete eradication.

However, different spartina elimination methods may have different effects on different groups of birds.

The Dafeng Milu National Nature Reserve (DMNNR), a critical coastal wetland in the Yellow Sea of China, was included in the Ramsar Convention's List of Wetlands of International Importance in 2002 and added to the Asia–Australasia bird migration protection network in 2003 [11]. The annual use of coastal wetlands in the DMNNR by thousands of migratory waterbirds indicates that these habitats are an important stopover as well as wintering and breeding sites for birds migrating from Australia to Siberia [12]. During the past few decades, the DMNNR wetlands have been subjected to loss and deterioration caused by the invasion of spartina which has gradually replaced native plant communities (common reed and *Suaeda salsa*) [13].

Owing to an increase in the abundance of the large herbivore milu (*Elaphures davidianus*) (hereafter elk) and ecological hydrological engineering, the area covered by spartina decreased significantly in the DMNNR [14]. At present, the impact of these alterations on avian diversity remains uncertain. In our study, we aimed to explore the impact of smooth cordgrass degradation on bird populations based on five years of monitoring of local avian diversity.

## 2. Materials and Methods

### 2.1. Study Area

The Dafeng Milu National Nature Reserve (DMNNR) (32°59′ N–33°03′ N, 120°47′ E–120°53′ E) is located in Jiangsu Province, China. The DMNNR consists of extensive areas previously covered by native vegetation and mudflats; these areas exhibit remarkable biodiversity and serve as suitable habitats for many wetland birds [13]. It was listed as a Ramsar wetland in 2002 and was added to the Asia–Australasia bird migration protection network in 2003 [12]. Due to intense anthropogenic economic activity and invasion by spartina, many areas along the Yellow Sea coastline have undergone significant alterations to their original natural features [15]. The study site was situated in the core area of the reserve. This area is characterized by minimal anthropogenic disturbances, and therefore has the exceptional natural landscape of the Yellow Sea tidal flat preserved. Historically, the vegetation composition within the reserve has primarily comprised common reed and *Suaeda salsa*. Following the invasion by smooth cordgrass, its range progressively expanded, swiftly displacing the native flora, and establishing a contiguous belt of monoculture vegetation along the shoreline [13]. Currently, after ecological hydrological engineering and an increase in elk numbers, the spartina area in the DMNNR has decreased significantly [14]. (Figure 1).

### 2.2. Field Surveys

Field work was carried out monthly from 2017 to 2021 in the core area of the DMNNR. Six sampling points with a 1 km radius were randomly established at the sites (Figure 1). To exclude year-to-year variation in bird patterns, we recorded data at the same sampling site by the same method every year. Bird surveys were conducted after high tide for 2 h, and they were performed by walking along canals as it was difficult to walk on the muddy intertidal flats and dense vegetation zones. At least two investigators conducted bird surveys using 8X binoculars (Nikon PROSTAFF 3S 8×42, Nikon Corporation, Tokyo, Japan) and a digital camera (Canon 7Dmark II and EF100–400 mm IS USM II, Canon, Tokyo, Japan). The species, abundance, and habitat types of birds seen and heard were recorded. All recorded species and the maximum counts within a month were utilized for analysis [16]. During our surveys, aerial foragers, such as Barn Swallows (*Hirundo rustica*), frequently engaged in low-altitude flight patterns above the vegetation layer and actively preyed upon airborne insects and these were recorded. Those birds that merely traversed the area by flight were excluded [4].

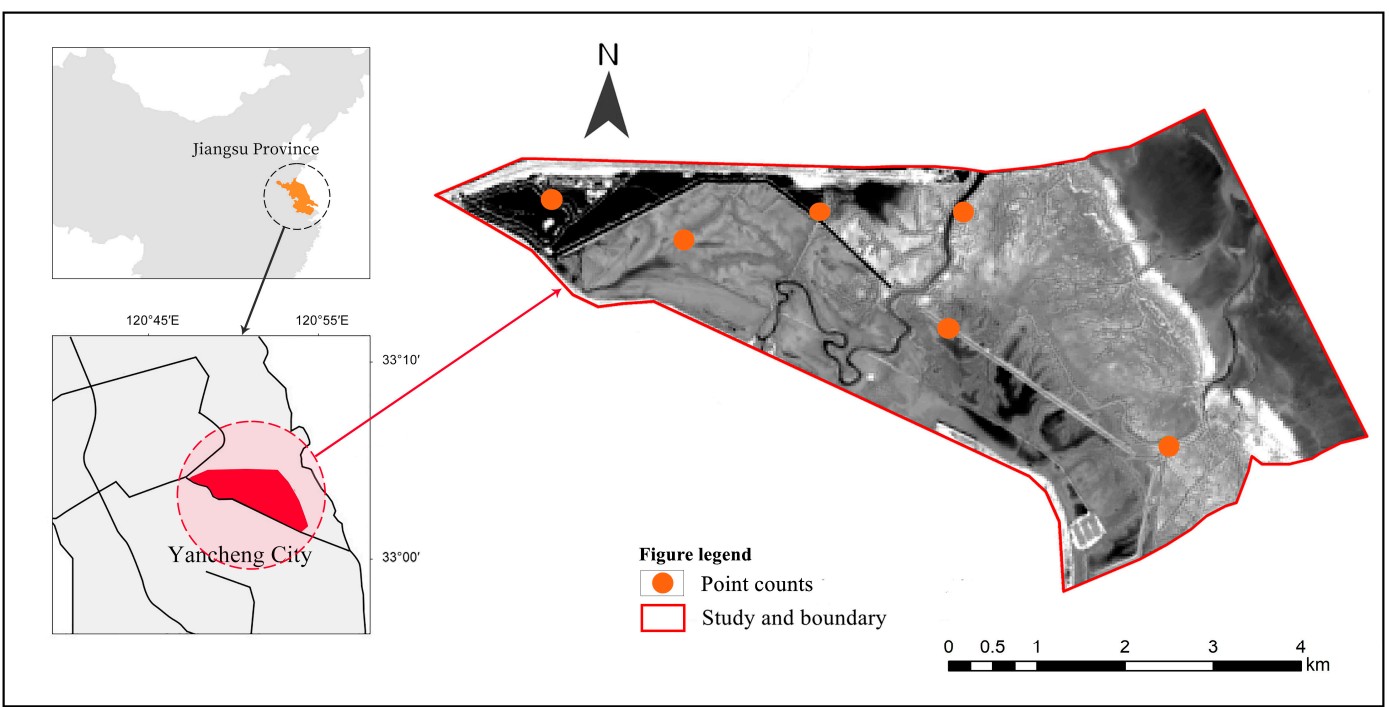

**Figure 1.** Study site in the Yancheng coastal wetland, Jiangsu Province, China.

*2.3. Data Analyses*

The spatial distribution of spartina exhibits interannual variation due to the species' high invasion rate; therefore, it is imperative to assess the annual changes in the extent of their distribution [14]. We utilized long time series Landsat imagery to monitor long-term changes in spartina populations. The study area was encompassed by a single path/row (P119R37) of the Landsat image [14]. Therefore, we selected all Landsat surface reflectance products with low cloud cover (<20%) captured during the summer period from 2017 to 2021, obtained through the GEE cloud computing platform. Finally, a collection of five Landsat8 images was acquired to cover the period from 2017 to 2021. The NDVI and modified normalized difference water index (MNDWI) were computed for each image collection and integrated into the corresponding collected images.

We classified the avian species into two groups: songbirds (including Passeriformes such as buntings, and parrotbills) and waterbirds (including herons, spoonbills, ducks, rails, shorebirds, gulls, and terns). Additionally, we differentiated them based on their resident status: breeding birds (breeding in DMNNR) and migrants (not breeding in DMNNR) [17]. Unidentifiable species were excluded from the estimation of species richness, so all the records were relatively conservative. The species richness and abundance of songbirds, waterbirds, breeding birds, and migrants were collected separately.

We used accumulation curves to calculate estimated species richness for each assemblage by time combination, as a means of reducing the bias of differing years [18]. The software package EstimateS 8.0 was used to construct randomized sample-based species accumulation curves for the observed species richness [19]. LSD-planned comparison tests were used to examine the differences in the observed species richness of total birds, songbirds, waterbirds, breeding birds, and migrants over 5 years.

Samples from the same year and bird groups were combined and we calculated the density of birds as D = N/S, where D is the density of birds (number per unit area), N is the number of birds recorded during the 12 months, and S is the area of the sampling site. The bird density was reported as the number of birds per hectare. The differences in the densities of total birds, songbirds, waterbirds, breeding birds, and migrants were further analyzed using repeated-measures ANOVAs, with sampling time as a within-subject factor and different years as a between-subject factor [20]. Tukey's honestly significant differ-

ence (HSD) tests were employed for conducting post hoc comparisons when statistically significant differences were observed. Before the analyses, all the data were log (n + 1) transformed to meet the assumptions required by ANOVAs [4]. The significance level of $p < 0.05$ was used for all the statistical tests, and the results are presented as means $\pm$ SE. The statistical software SPSS 19.0 (IBM Corporation, New York, NY, America) was used for all the analyses.

To investigate the relationship between birds and habitat types in the DMNNR, correspondence analysis (CA) was performed [16]. Species with more than 100 individuals counted in at least one survey were included in the correspondence analysis [21]. These species were classified into four groups: swan, goose, and duck (Anseriformes); shorebird (Charadriiformes); songbird (Passeriformes) and others (i.e., all species not applicable to the former three groups including species from the families of Podicipediformes, Ciconiiformes and Gruiformes). We conducted an analysis for the four groups using R (4.0.1).

## 3. Results

### 3.1. Changes in Area Covered by Spartina from 2017 to 2021

A map of the annual habitat change for 2017–2021 shows the area and distribution of spartina (Figure 2). The annual area of this species in the DMNNR was modeled by linear regression. The area of spartina decreased significantly in the study area at a rate of 310 ha per year and by 62% during 2017–2021 ($p < 0.01$) (Figure 3). The extent of the mudflat in the summer of 2017–2018 is so limited that it is not shown on the map. Mudflats began to appear in 2019, and the area of bare land gradually increased (Figure 2). In 2017–2019, the water area maintained a high coverage rate, and in 2020–2021, the coverage rate decreased rapidly (Figure 2). The area of spartina closer to the ocean maintained a high density during 2017–2021 (Figure 2).

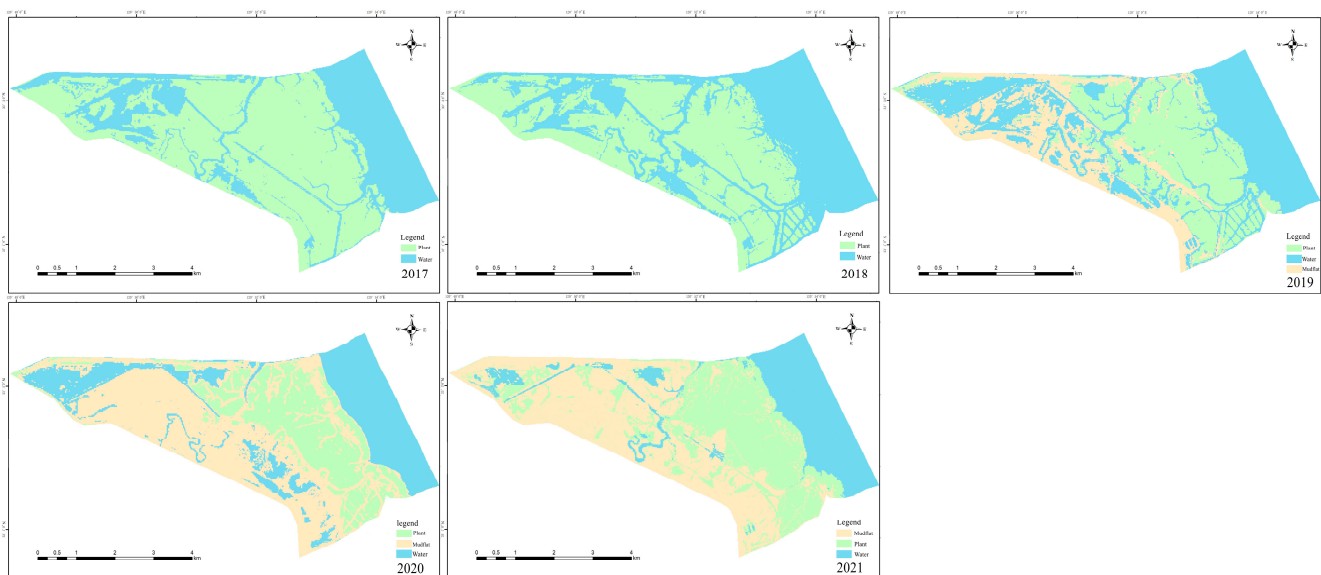

**Figure 2.** Vegetation coverage change during 2017–2021 in the DMNNR.

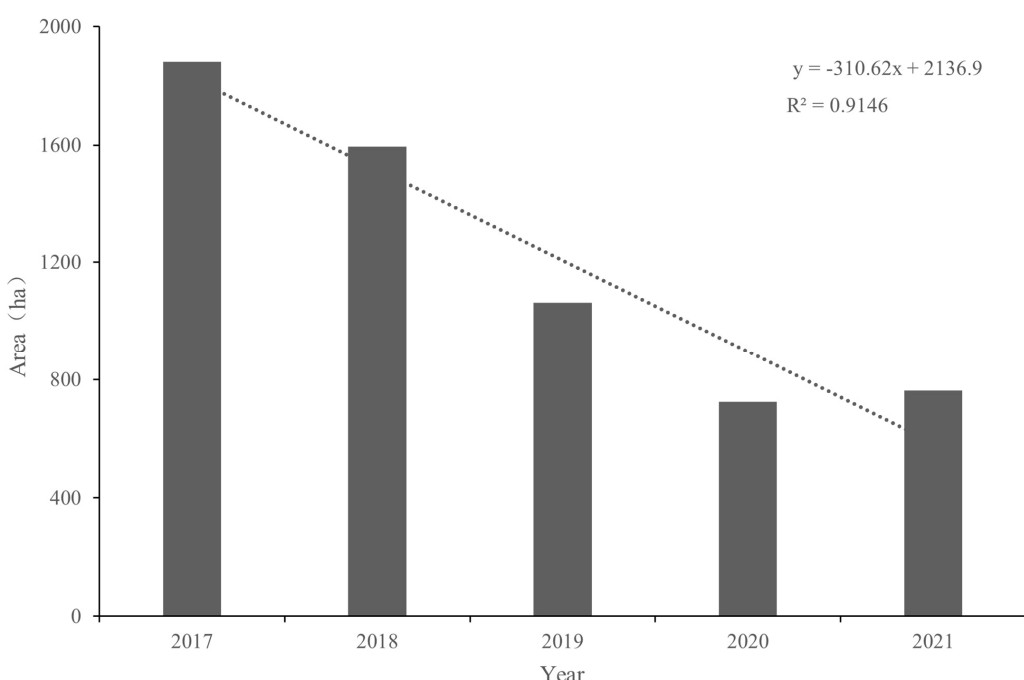

**Figure 3.** The spartina area in the DMNNR was modeled via linear regression during 2017–2021.

*3.2. Bird Species*

From 2017 to 2021, a total of 124 bird species were identified and belonged to 10 orders and 36 families (see Appendix A). Species number varied among different years: the highest number was recorded in 2019 (116 species), followed by 2018 (111 species). Fewer species were recorded in 2017 (101 species), 2020 (97 species) and 2021 (90 species) (Figure 4). In 2019, the species richness of total bird species was greater than in the other years, and in 2020 and 2021, the number of bird species was significantly lower than that in the other years (LSD-planned comparison test, $p < 0.01$ for all).

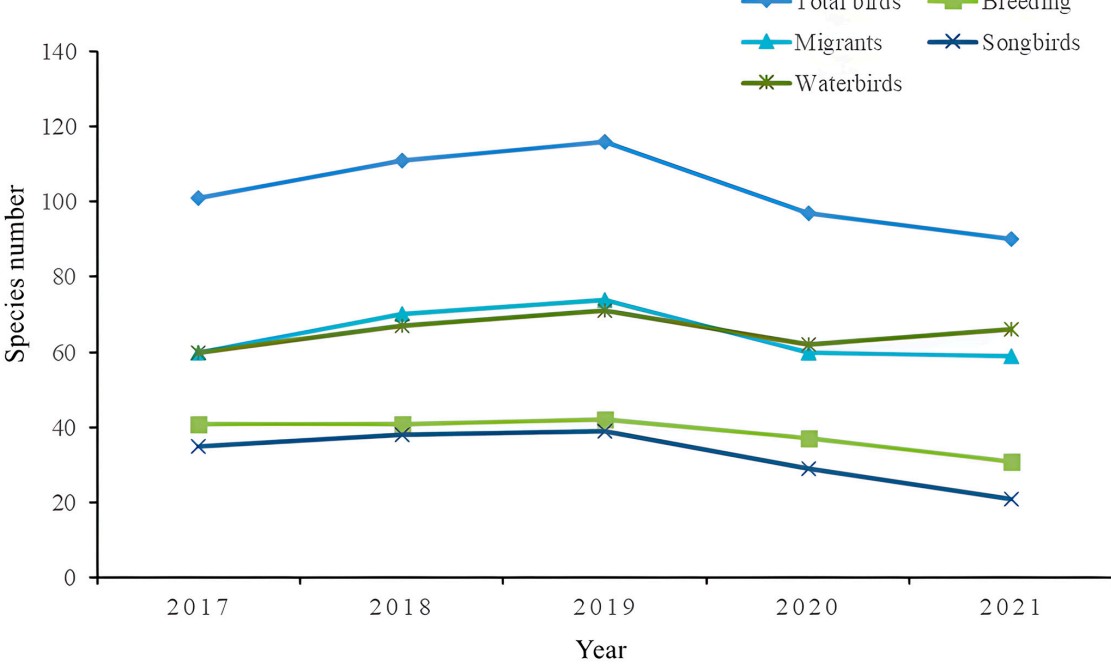

**Figure 4.** Species richness of birds in different groups observed over 5 years, from 2017 to 2021, in the DMNNR.

From 2017 to 2021, most of the species were found to be migrants (82 species), and the others (42 species) were breeding birds. More migrant species were recorded in 2019 (74 species) than in other years ($\leq$70 species). In 2018 and 2019, the observed species richness of migrant species was much greater than that in the other years (2017, 2020 and 2021) (LSD-planned comparison test, $p < 0.01$ for all). Most breeding species were recorded in 2019 (42 species), while very few breeding species were recorded in 2021 (31 species) and 2020 (37 species).

Waterbirds are the most important part of the bird composition of the reserve. A total of 78 species of waterbirds were recorded over five years. Most waterbird species were recorded in 2019 (74 species) compared to the other years (68 species). The number of waterbird species observed was significantly greater in 2019 (74) and 2018 (70) than in 2021 (68), 2020 (65) and 2017 (63) (all $p < 0.01$). Most songbirds were also recorded in 2019 (42 species), while very few songbirds were recorded in 2021 (22 species), 2020 (32 species), and 2017 (38 species). In addition, the number of observed species of songbirds was significantly greater in 2019 (42) than in 2020, 2017 and 2021 (all $p < 0.01$) and was not significantly different from that in 2018.

### 3.3. Bird Abundance and Density

A total of 65,658 birds were recorded during the 5 years; 84.1% (55,244 birds) were waterbirds and 13.9% (10,414 birds) were songbirds. Most birds (26.9% of the total) were recorded in 2019, with 20.0% being found in 2021, 19.6% in 2020, 19.5% in 2018 and 14.1% in 2017. Waterbirds constituted the predominant avian species during the 5 years (73.6% in 2017, 72.3% in 2018, 83.0% in 2019, 93.8% in 2020 and 95.2% in 2021), while songbirds were rare in 2021 (623 of 13,047 birds), 2020 (794 of 12,867 birds), 2017 (2439 of 9247 birds), 2019 (3010 of 17,693 birds), and 2018 (3548 of 12,804 birds). In terms of resident status, 70.4% of all birds were migrants, with an overwhelming majority of migrants (70.6% of total migrants) being recorded from 2019 to 2021 and only 29.4% being recorded in 2017 and 2018. Among the breeding birds, most were found in 2019 (5691 individuals, 29.2% of the total), 25.0% (4872 individuals) in 2018, 18.4% (3578 individuals) in 2017, 13.9% (2703 individuals) in 2021, and 13.4% (2616 individuals) in 2020.

Most birds observed in all of the years were shorebirds (17,916 individuals, 27.3% of the total); pied avocet (*Himantopus himantopus*) was the most abundant shorebird (4935 birds, 27.5% of the total) and was recorded the most in 2019. The second most abundant bird was Dunlin (*Calidris alpina*, 2749 birds and 15.3% of the total), which was the dominant species in 2021. The vinous-throated parrotbill (*Paradoxornis webbianus*) (1563 records) was the numerically dominant species among the songbirds, accounting for 15.0% of the total songbirds recorded there.

Comparisons indicated that the average density of total birds was significantly lower in 2017 (3.14 $\pm$ 0.04 ind/ha) than in 2019 (6.00 $\pm$ 0.06 ind/ha), while there was no significant difference in the average density of total birds among the other years ($p > 0.05$ for all). The average density of songbirds was significantly lower in 2021 (0.21 $\pm$ 0.01 ind/ha) than in 2018 (1.20 $\pm$ 0.58 ind/ha), 2019 (1.02 $\pm$ 0.03 ind/ha), and 2017 (0.83 $\pm$ 0.02 ind/ha) ($p < 0.001$), while there was no significant difference in 2021 (0.21 $\pm$ 0.01 ind/ha ($p > 0.05$ for both)). Waterbird densities were significantly lower in 2017 (2.31 $\pm$ 0.05 ind/ha) and 2018 (3.14 $\pm$ 0.07 ind/ha) than in 2019 (4.98 $\pm$ 0.09 ind/ha) ($p < 0.001$); however, there was no significant difference in waterbird density between 2018, 2020 and 2021 ($p > 0.05$ for all) (Figure 5).

### 3.4. Spatial Variations of Bird Communities

Based on the criteria specified (see Section 2), 31 species (Appendix B) were selected for the analysis of habitat use. The results of the correspondence analysis are presented in Figure 6. It was clear that the different groups of birds used different habitat types. Swans, geese, and ducks were predominantly distributed in open water, followed by rivers and mudflats (see Figure 6a). The shorebirds mainly used open water, followed by mudflats (see Figure 6b). Most of the Songbirds were observed in common reed areas, followed by spartina areas and

mudflat (see Figure 6c). Other groups, such as common coots, were mainly observed in open water, while gray heron was primarily distributed in mudflats (see Figure 6d).

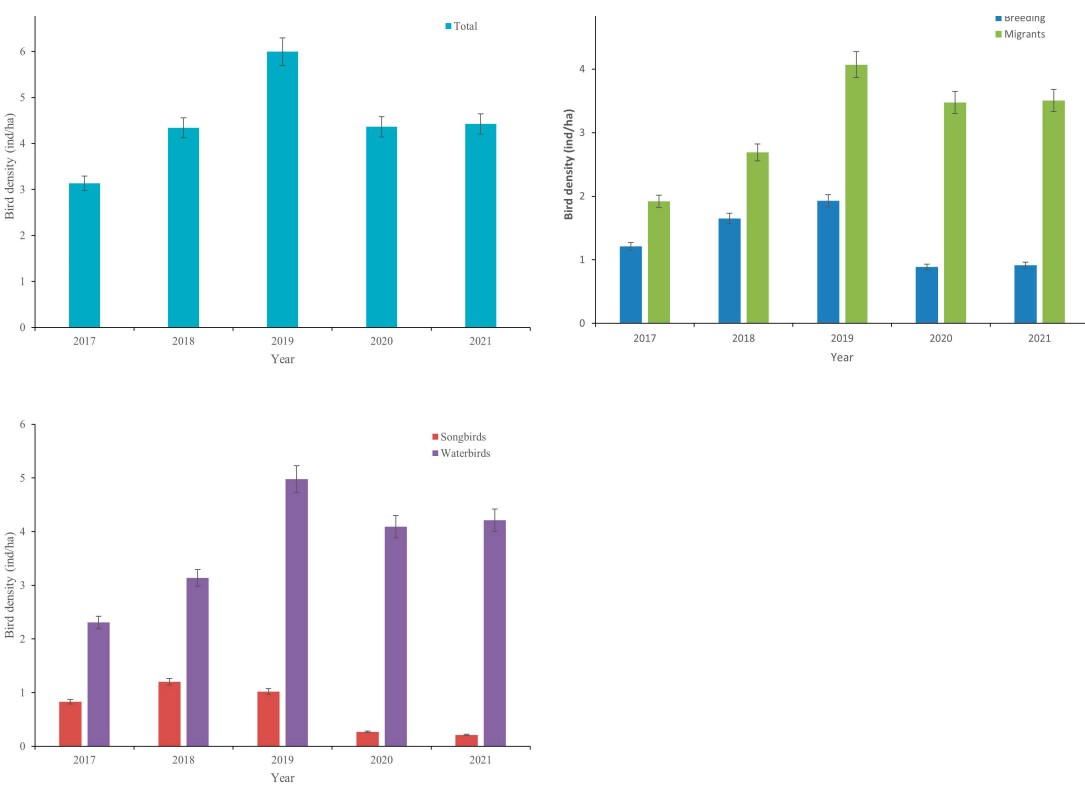

**Figure 5.** Mean bird densities (±SE) of various bird groups in the five years.

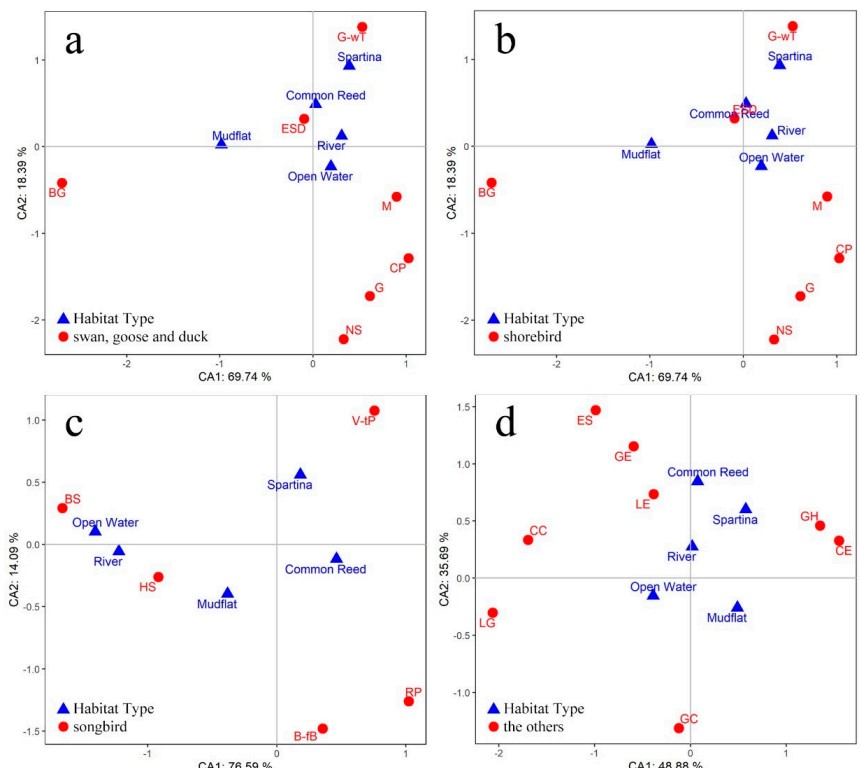

**Figure 6.** Correspondence analysis of habitat types and bird species in the DMNNR. The regular triangles represent habitat types; the solid circles represent different bird species. Waterbird species were classified into four groups: (**a**) swan, goose, and duck; (**b**) shorebird; (**c**) songbird; and (**d**) others.

## 4. Discussion

The annual area of *S. alterniflora* decreased substantially in the study area, which might be attributable to the foraging of elks and its trampling of spartina [14,22]. With the number of elk individuals in the study area increasing from approximately 846 individuals in 2017 to more than 2658 in 2021, the number of elks foraging on spartina has increased greatly. The construction of freshwater artificial ditches for elk drinking also resulted in a subsequent downward trend in the spartina area in the study area. Although the annual change in NDVI is commonly used to depict the vegetation sequence and indicate trends of decrease or increase, its accuracy is not entirely reliable [23,24]. The algorithm's selection of low-cloud images for cloud removal may result in missing images, thereby impacting image combination, and potentially leading to classification errors. For instance, due to the limited availability of cloud-free pixels in the July and August months within the spartina dataset, the NDVI values of these pixels in the most verdant image might be lower than their maximum value, potentially leading to misclassification of spartina areas as other saltmarshes in the annual maps [14]. Therefore, we systematically constructed annual maps of spartina coverage at regular intervals spanning multiple years to address and minimize potential uncertainties. The spatial and temporal patterns of spartina communities exhibit a robust correlation with hydrological and soil environmental factors [25]. The research in the future could investigate the distribution pattern and evolutionary trends of spartina in response to hydrological and soil environmental stressors.

The impacts of various spartina eradication methods on avian species vary. In California, a small number of native birds also enter spartina areas to breed, and rapid removal of invasive plants may have a negative impact on these breeding birds [11]. The number of shorebirds, geese, and ducks in the Shanghai Chongming Dongtan wetland in China has been effectively restored through manual removal and waterlogging of spartina [10]. In our study, progressive degeneration of spartina occurred after ecological hydrological engineering and an increase in elk numbers. The bird species richness and abundance exhibited a significant increase in 2019. However, as vegetation degradation intensified, bird abundance and species richness declined in 2020 and 2021. The main reasons may be that the dense and homogeneous vegetation community structure of spartina is not conducive to providing a bird habitat or a lack of adequate food resources [3,9]. A reduction in vegetation coverage coupled with an increase in habitat heterogeneity results in elevated species richness and bird abundance.

Different habitats provide different functional services for birds [16]. The invasion of spartina may provide vegetation conditions similar to those of local habitats for passerine birds, and habitat changes caused by invasive plants may drive the adaptive evolution of native animal behavior [6]. A dense vegetation structure is beneficial for small birds hiding nest sites and avoiding predators but unfavorable for large birds that need open land [16,26]. In our study, a decrease in vegetation coverage led to a severe decline in songbirds and an increase in waterbirds in open water and mudflats. The breeding birds mainly comprises songbirds (such as *Sinosuthora webbiana*) that breed in spartina and phragmites habitats, as well as a limited number of ground-nesting birds such as Vanellus cinereus. Most breeding songbirds are negatively affected by vegetation degradation. The habitat of ground-nesting birds increased; however, their population remained stable. It may be that the number of elk was too large and frequent activity would increase the risk of tramping on bird nests on the ground. The primary avian migrants comprise wetland waterbirds, and the expansion of mudflat and water areas has resulted in an increase in migrant populations. Therefore, maintaining suitable habitat heterogeneity is critical for maintaining bird diversity.

## 5. Conclusions

The richness of local bird species decreased with spartina degradation, and the breeding passerines ultimately disappeared from the reserve. Although the richness and abundance of waterbirds increased during 2017–2021, with the continuous degradation of vegetation and soil erosion, the number of waterbirds may decrease in the future. Given the

growing threats to wetland ecosystems from elk, we recommend that further research be conducted to ensure a balance between the number of elks and bird diversity conservation. It is recommended that native vegetation, such as reeds and Suaeda, be restored to the reserve as soon as possible and that the elk population be controlled within a reasonable range.

**Author Contributions:** Conceptualization, T.C. and C.L.; methodology, T.C., P.C. and B.L.; formal analysis, T.C. and D.W.; writing—original draft preparation, T.C. and D.W.; writing—review and editing, T.C. and D.W. All authors have read and agreed to the published version of the manuscript.

**Funding:** This work is supported by the National Natural Science Foundation of China (32171526).

**Institutional Review Board Statement:** Not applicable.

**Data Availability Statement:** The data presented in this study are available in this published article.

**Acknowledgments:** We thank patrols at the Dafeng Milu National Nature Reserve for assistance and support in the field sampling.

**Conflicts of Interest:** The authors declare that they have no known competing financial interests or personal relationships that could have appeared to influence the work reported in this paper.

## Appendix A. The Average Number of Birds during 2017 to 2021 in DMNNR

**Table A1.** The average number of birds during 2017 to 2021 in DMNNR.

| Common Name | Scientific Name | IUCN | 2017 (n = 12) | 2018 (n = 12) | 2019 (n = 12) | 2020 (n = 12) | 2021 (n = 12) |
|---|---|---|---|---|---|---|---|
| Little Grebe | *Tachybaptus ruficollis* | | 14.4 | 13.6 | 17.0 | 4.8 | 5.9 |
| Great Crested Grebe | *Podiceps cristatus* | | 0.3 | 0.2 | | | |
| Great Cormorant | *Phalacrocorax carbo* | | 6.9 | 30.9 | 107.3 | 228.8 | 223.5 |
| Grey Heron | *Ardea cinerea* | | 34.0 | 46.9 | 69.9 | 112.8 | 103.6 |
| Great Egret | *Egretta alba* | | 15.2 | 23.9 | 40.9 | 57.5 | 36.5 |
| Intermediate Egret | *Egretta intermedia* | | 12.2 | 16.5 | 23.7 | 16.9 | 19.8 |
| Little Egret | *Egretta garzetta* | | 23.2 | 37.2 | 66.8 | 55.3 | 37.9 |
| Cattle Egret | *Bubulcus ibis* | | 26.5 | 29.4 | 31.3 | 19.8 | 4.4 |
| Chinese Pond Heron | *Ardeola bacchus* | | 2.0 | 5.0 | 5.1 | 0.8 | |
| Striated Heron | *Butorides striata* | | 1.8 | 0.8 | 0.3 | | |
| Black-crowned Night Heron | *Nycticorax nycticorax* | | 5.3 | 6.3 | 15.5 | 2.6 | 3.2 |
| Eurasian Bittern | *Botaurus stellaris* | | 0.8 | 0.3 | 0.3 | | |
| Oriental Stork | *Ciconia boyciana* | EN | 3.2 | 13.8 | 15.8 | 2.4 | 1.5 |
| Eurasian Spoonbill | *Platalea leucorodia* | | 17.1 | 10.2 | 28.0 | 24.5 | 28.2 |
| Black-faced Spoonbill | *Platalea minor* | EN | 2.6 | 1.2 | 1.6 | 3.3 | 2.9 |
| Bean Goose | *Anser fabalis* | | 12.7 | 11.9 | 45.5 | 5.4 | 37.6 |
| Bean Goose | *Anser albifrons* | | 0.9 | 3.8 | 0.7 | | 1.8 |
| Lesser White-fronted Goose | *Anser erythropus* | VU | 0.3 | | 0.3 | | 0.8 |
| Common Shelduck | *Tadorna tadorna* | | | | | 9.2 | |
| Gadwall | *Anas strepera* | | 9.2 | 20.1 | 24.5 | 5.8 | 1.6 |
| Falcated Duck | *Anas falcata* | NT | 3.7 | | 4.7 | | 0.8 |
| Eurasian Wigeon | *Anas penelope* | | 5.3 | 2.5 | 1.3 | 9.9 | 11.7 |
| Mallard | *Anas platyrhynchos* | | 25.7 | 2.8 | 45.3 | 28.6 | 6.8 |
| Eastern Spot-billed Duck | *Anas poecilorhyncha* | | 95.1 | 92.8 | 186.7 | 120.6 | 44.0 |
| Northern Shoveler | *Anas clypeata* | | 1.8 | 2.4 | 25.8 | 0.5 | 16.8 |
| Northern Pintail | *Anas acuta* | | 2.6 | 0.9 | 2.3 | 4.5 | |
| Garganey | *Anas querquedula* | | 1.7 | 0.6 | 3.9 | | 1.9 |
| Green-winged Teal | *Anas crecca* | | 44.7 | 38.8 | 54.2 | 47.0 | 23.3 |
| Common Pochard | *Aythya ferina* | VU | 35.9 | 14.7 | 15.5 | 6.3 | 1.8 |
| Ferruginous Duck | *Aythya nyroca* | NT | 11.9 | 1.2 | | | |
| Tufted Duck | *Aythya fuligula* | | 13.6 | 9.5 | 4.2 | 2.5 | |
| Greater Scaup | *Aythya marila* | | 2.3 | 1.2 | 0.4 | | 2.9 |
| Smew | *Mergellus albellus* | | 1.0 | | | | 1.5 |
| Common Pheasant | *Phasianus colchicus* | | 5.5 | 1.7 | 0.8 | 0.3 | |
| Common Crane | *Grus grus* | | | 0.3 | | | |
| Red-crowned Crane | *Grus japonensis* | EN | 0.9 | 0.3 | | 0.3 | 0.3 |
| Common Moorhen | *Gallinula chloropus* | | 7.3 | 3.8 | 1.5 | 1.9 | 0.4 |
| Common Coot | *Fulica atra* | | 17.7 | 17.8 | 24.8 | 20.6 | 10.7 |
| Black-winged Stilt | *Himantopus himantopus* | | 3.3 | 3.0 | 5.5 | 15.5 | 3.7 |
| Pied Avocet | *Himantopus himantopus* | | 18.0 | 70.4 | 93.2 | 228.3 | 77.5 |
| Oriental Pratincole | *Glareolidae* | | 6.7 | 8.7 | 3.5 | 1.6 | 0.8 |
| Northern Lapwing | *Vanellus vanellus* | NT | 6.2 | 7.3 | 34.2 | 10.9 | 4.8 |
| Grey-headed Lapwing | *Vanellus cinereus* | | 16.9 | 11.2 | 12.7 | 24.5 | 22.7 |
| Pacific Golden Plover | *Pluvialis fulva* | | 0.3 | 1.0 | 0.5 | | 3.2 |

**Table A1.** *Cont.*

| Common Name | Scientific Name | IUCN | 2017 (n = 12) | 2018 (n = 12) | 2019 (n = 12) | 2020 (n = 12) | 2021 (n = 12) |
|---|---|---|---|---|---|---|---|
| Grey Plover | *Pluvialis squatarola* | | 0.7 | 1.9 | 0.3 | 3.0 | 4.4 |
| Little Ringed Plover | *Charadrius dubius* | | 3.9 | 3.0 | 5.3 | 3.6 | 3.3 |
| Kentish Plover | *Charadrius dubius* | | 6.5 | 9.0 | 18.9 | 27.0 | 14.3 |
| Lesser Sand Plover | *Charadrius mongolus* | | 0.1 | 2.3 | 0.9 | 0.8 | 7.8 |
| Greater Sand Plover | *Charadrius leschenaultii* | | 0.9 | 2.3 | 11.3 | 1.8 | 7.9 |
| Common Snipe | *Gallinago gallinago* | | 2.8 | 2.3 | 0.2 | 1.0 | 0.2 |
| Long-billed Dowitcher | *Limnodromus scolopaceus* | | | | 0.1 | 0.1 | |
| Black-tailed Godwit | *Limosa limosa* | | 2.2 | 61.7 | 30.4 | 16.3 | 12.2 |
| Bar-tailed Godwit | *Limosa lapponica* | | 4.2 | 5.3 | 7.8 | 2.6 | 6.5 |
| Eurasian Curlew | *Numenius arquata* | NT | 9.8 | 2.1 | 3.4 | 27.0 | 13.3 |
| Eastern Curlew | *Numenius madagascariensis* | EN | 1.8 | 0.8 | 0.9 | 2.0 | 1.1 |
| Spotted Redshank | *Tringa erythropus* | | 6.1 | 6.1 | 10.8 | 3.5 | 4.4 |
| Common Redshank | *Tringa totanus* | | 10.0 | 12.2 | 9.4 | 5.9 | 11.4 |
| Marsh Sandpiper | *Tringa stagnatilis* | | 2.2 | 1.3 | 1.3 | 16.1 | 2.5 |
| Common Greenshank | *Tringa nebularia* | | 5.3 | 4.5 | 5.7 | 19.5 | 5.7 |
| Green Sandpiper | *Tringa ochropus* | | 0.1 | 0.1 | 0.8 | 0.8 | 0.5 |
| Wood Sandpiper | *Tringa glareola* | | 0.1 | 0.1 | 0.1 | 0.1 | 0.3 |
| Common Sandpiper | *Actitis hypoleucos* | | 1.4 | 0.4 | 0.3 | 3.3 | 0.7 |
| Grey-tailed Tattler | *Heteroscelus brevipes* | | | | 0.5 | 0.8 | 1.0 |
| Sanderling | *Calidris alba* | | | | | 33.8 | 12.5 |
| Red-necked Stint | *Calidris ruficollis* | | | | 5.0 | 165.0 | 38.3 |
| Sharp-tailed Sandpiper | *Calidris acuminata* | | 15.3 | 10.7 | 19.8 | 85.0 | 88.3 |
| Dunlin | *Calidris alpina* | | 8.7 | 19.4 | 45.0 | 67.8 | 152.5 |
| Curlew Sandpiper | *Calidris ferruginea* | | | | | 0.8 | 0.2 |
| Broad-billed Sandpiper | *Limicola falcinellus* | | | | | 0.5 | 0.3 |
| Eurasian Oystercatcher | *Haematopus ostralegus* | | 0.7 | | 0.3 | | |
| Black-tailed Gull | *Larus crassirostris* | | 0.4 | 0.4 | 0.3 | 0.4 | 0.9 |
| Caspian Gull | *Larus cachinnans* | | 0.3 | 1.7 | 1.7 | 2.3 | 6.6 |
| Slaty-backed Gull | *Larus schistisagus* | | 0.1 | 0.6 | 14.0 | 1.4 | 0.5 |
| Black-headed Gull | *Chroicocephalus ridibundus* | | 18.0 | 18.4 | 10.8 | 67.9 | 21.6 |
| Saunders's Gull | *Saundersilarus saundersi* | VU | 5.0 | 8.6 | 25.9 | 4.4 | 29.7 |
| Gull-billed Tern | *Gelochelidon nilotica* | | 1.7 | | 12.5 | | 1.0 |
| Common Tern | *Sterna hirundo* | | 16.8 | 21.8 | 11.9 | 5.5 | 7.8 |
| Little Tern | *Sternula albifrons* | | 16.3 | 12.3 | 30.8 | 19.8 | 8.9 |
| White-winged Tern | *Chlidonias leucoptera* | | 12.8 | 1.7 | | | |
| Common Cuckoo | *Cuculus canorus* | | 1.8 | 1.5 | 1.3 | 1.4 | 0.4 |
| Lesser Coucal | *Centropus toulou* | | 1.3 | 1.2 | 0.5 | 0.1 | |
| Common Kingfisher | *Alcedo atthis* | | 2.0 | 1.4 | 0.3 | 0.3 | 0.2 |
| Pied Kingfisher | *Ceryle rudis* | | 1.1 | 1.8 | 1.0 | 0.5 | 0.2 |
| Common Hoopoe | *Upupa epops* | | 2.2 | 1.8 | 1.4 | 0.5 | 0.4 |
| Oriental Skylark | *Alauda gulgula* | | 2.2 | 1.8 | 2.7 | 0.3 | 1.0 |
| Barn Swallow | *Hirundo rustica* | | 15.4 | 14.8 | 11.0 | 11.1 | 17.3 |
| Red-rumped Swallow | *Cecropis daurica* | | 7.1 | 7.8 | 9.5 | 9.8 | 10.4 |
| White Wagtail | *Motacilla alba* | | 2.1 | 0.8 | 1.0 | 2.0 | 0.5 |
| Gray Wagtail | *Motacilla cinerea* | | 3.5 | 1.3 | 0.3 | 0.5 | 0.2 |
| Olive-backed Pipit | *Anthus hodgsoni* | | 1.1 | 0.5 | | | |
| Light-vented Bulbul | *Pycnonotus sinensis* | | 6.3 | 5.5 | 6.7 | 4.1 | 3.5 |
| Tiger Shrike | *Lanius tigrinus* | | 0.5 | 0.1 | | | |
| Long-tailed Shrike | *Lanius schach* | | 4.1 | 4.2 | 2.6 | 1.5 | 1.2 |
| Brown Shrike | *Lanius cristatus* | | 2.0 | 1.8 | 2.9 | 0.6 | 0.3 |
| Black Drongo | *Dicrurus macrocercus* | | 9.1 | 9.5 | 11.1 | 5.5 | 3.6 |
| White-cheeked Starling | *Sturnus cineraceus* | | 9.6 | 12.5 | 11.3 | 8.8 | 1.9 |
| Silky Starling | *Spodiopsar sericeus* | | 11.4 | 5.8 | 10.1 | 6.8 | 0.3 |
| Common Magpie | *Pica Pica* | | 8.6 | 9.8 | 10.3 | 3.4 | 1.7 |
| Orange-flanked Bluetail | *Tarsiger cyanurus* | | 0.8 | 0.3 | 0.3 | | |
| Daurian Redstart | *Phoenicurus auroreus* | | 3.5 | 1.7 | 1.7 | | |
| Siberian Thrush | *Geokichla sibirica* | | 0.3 | | | | |
| Dusky Thrush | *Turdus eunomus* | | 0.9 | 0.7 | 3.0 | | |
| Grey-streaked Flycatcher | *Muscicapa griseisticta* | | 1.3 | 0.8 | 1.3 | | |
| Mugimaki Flycatcher | *Ficedula mugimaki* | | 0.5 | 0.3 | | | |
| Vinous-throated Parrotbill | *Paradoxornis webbianus* | NT | 43.8 | 52.8 | 40.3 | 10.5 | 3.0 |
| Reed Parrotbill | *Paradoxornis heudei* | | 12.8 | 13.5 | 17.0 | 2.0 | |
| Zitting Cisticola | *Cisticola juncidis* | | 17.6 | 10.8 | 7.1 | 2.8 | 0.2 |
| Plain Prinia | *Prinia inornata* | | 14.2 | 11.6 | 9.5 | 3.3 | 0.5 |
| Oriental Reed Warbler | *Acrocephalus orientalis* | | 2.8 | 4.2 | 4.3 | 1.3 | |
| Black-browed Reed Warbler | *Acrocephalus bistrigiceps* | | | 1.5 | 2.6 | | |
| Yellow-browed Warbler | *Phylloscopus inornatus* | | 0.3 | 0.2 | 0.7 | | |
| Marsh Grassbird | *Locustella pryeri* | | 14.0 | 5.1 | 1.9 | | |
| Chinese Penduline Tit | *Remiz consobrinus* | | 4.8 | 5.2 | 2.3 | 1.6 | |
| Cinereous Tit | *Parus major* | | 3.5 | 3.3 | 3.1 | 1.3 | 0.2 |
| House Sparrow | *Passer montanus* | | 38.8 | 27.8 | 11.5 | 7.1 | 2.9 |
| Brambling | *Fringilla montifringilla* | | 3.8 | 3.2 | | | |
| Meadow Bunting | *Emberiza ioides* | | 4.6 | 8.7 | 6.7 | 1.3 | |
| Chestnut-eared Bunting | *Emberiza fucata* | | 3.2 | 1.8 | 0.9 | 0.1 | |
| Little Bunting | *Emberiza pusilla* | | 7.0 | 8.8 | 8.2 | 2.3 | |
| Rustic Bunting | *Emberiza rustica* | | 8.2 | 12.6 | 10.8 | 1.3 | 0.8 |
| Chestnut Bunting | *Emberiza rutila* | | 0.5 | 0.3 | 0.3 | | |
| Black-faced Bunting | *Emberiza spodocephala* | | 19.4 | 29.2 | 15.8 | 4.8 | 0.3 |
| Pallas's Bunting | *Emberiza pallasi* | | 17.5 | 10.3 | 6.8 | 1.5 | 0.9 |
| Reed Bunting | *Emberiza schoeniclus* | | 4.7 | 0.8 | 0.3 | | |

## Appendix B. The Average Bird Number ≥ 100 Individuals during 2017 to 2021 in Different Groups

**Table A2.** The average bird number ≥ 100 individuals during 2017 to 2021 in different groups.

| Group | Common Name | Abbreviation | River | Open Water | Mudflat | Common Reed | Spartina | Sum |
|---|---|---|---|---|---|---|---|---|
| Swan, goose, and duck | Bean Goose | BG | 17 | 55 | 175 | | | 247 |
| | Gadwall | G | 42 | 90 | 10 | | | 142 |
| | Mallard | M | 105 | 120 | 7 | 2 | 5 | 239 |
| | Eastern Spot-billed Duck | ESD | 300 | 380 | 200 | 100 | 17 | 997 |
| | Northern Shoveler | NS | 20 | 80 | 13 | | | 113 |
| | Green-winged Teal | G-wT | 200 | 150 | 50 | 21 | 40 | 461 |
| | Common Pochard | CP | 73 | 100 | | | | 173 |
| Shorebird | Pied Avocet | PA | 266 | 439 | 252 | 30 | | 987 |
| | Northern Lapwing | NL | 23 | 7 | 101 | 12 | | 143 |
| | Grey-headed Lapwing | G-hL | 44 | 12 | 129 | 6 | | 191 |
| | Black-tailed Godwit | B-tG | 70 | 204 | 5 | 2 | | 281 |
| | Common Redshank | CR | 9 | 103 | | 1 | | 113 |
| | Sharp-tailed Sandpiper | S-tS | 27 | 120 | 105 | 30 | 180 | 458 |
| | Dunlin | D | 183 | 200 | 175 | 50 | | 550 |
| | Black-headed Gull | B-hG | 90 | 100 | 74 | 10 | | 274 |
| | Saunders's Gull | SG | 30 | 73 | 60 | 10 | | 173 |
| | Common Tern | CT | 40 | 90 | 19 | | | 149 |
| | Little Tern | LT | 66 | 110 | 20 | | | 196 |
| Songbird | Barn Swallow | BS | 45 | 50 | 18 | 25 | 20 | 158 |
| | Vinous-throated Parrotbill | V-tP | | | 13 | 200 | 100 | 313 |
| | Reed Parrotbill | RP | | | | 105 | 2 | 107 |
| | House Sparrow | HS | 28 | 26 | 30 | 40 | 22 | 146 |
| | Black-faced Bunting | B-fB | 7 | | 33 | 90 | 12 | 142 |
| Others | Little Grebe | LG | 34 | 82 | | | | 116 |
| | Great Cormorant | GC | 174 | 556 | 401 | | | 1131 |
| | Grey Heron | GH | 268 | 100 | 300 | 42 | 27 | 737 |
| | Great Egret | GE | 113 | 155 | 48 | 52 | 4 | 372 |
| | Little Egret | LE | 77 | 163 | 59 | 37 | 22 | 484 |
| | Cattle Egret | CE | 40 | 33 | 101 | 28 | 5 | 207 |
| | Eurasian Spoonbill | ES | 88 | 105 | 12 | 34 | | 219 |
| | Common Coot | CC | 79 | 120 | | 4 | | 203 |

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
