# Peer review of "Effects of Invasive Smooth Cordgrass Degradation on Avian Species Diversity in the Dafeng Milu National Nature Reserve, a Ramsar Wetland on the Eastern Coast of China"

_diversity, doi:10.3390/d16030176_

Round 1

Reviewer 1 Report

Comments and Suggestions for Authors

Well done research. Changes in the environment structure also cause changes in the species community. 

Author Response

Thank you very much for taking the time to review this manuscript.

Reviewer 2 Report

Comments and Suggestions for Authors

Scientific names of animals in plant cursive

Space before [

Line 12 - henceforth referred to as DMNNR, I think DMNNR is enough

Line 12 why elk instead of milu

Line 20 with the increasing

Abstract covers subject of the study quite well

Line 29 (spartina)

Line 36 Phragmites names start with upper or lower case, be consistent, habitats [

Introduction describes the basics quite well. I would like to read a more detailed research question.

Material and methods

Study area. This par could have a more logical order

Field survey – sample points were chosen randomly and sampled regularly

Fig. 1

Line 108 you can also reference to the appendix

Data analysis can be divided into spartina and birds

Rest of material and methods is fine

Results

Line 143-146 should be shifted to the discussion

Fig. 2 Legend is very small and nearly unreadable. You can use the space on the bottom right and use only one legend, one scale and one north arrow

Fig. 3 I miss significant level

Fig. 6 I miss descriptions of the axis and abbreviations of the habitat types

Discussion

NDVI ??

I would start with change in spartina and why and then discuss methods

The discussion is mainly descriptive and there is no interpretation why breeding birds and/or migrants are affected. This may be discussed more deeply.

Conclusion is descriptive only. The results are described again but why does it happen and what are the consequences for the future?

The study is well planned,, the results are well presented. The discussion should go into detail in some aspects and the conclusion is more a summary than a conclusion.

There are several formating issues

Author Response

Thank you very much for taking the time to review this manuscript.

Comments 1:why elk instead of milu

Response 1:Milu was Chinese phonetic alphabets, elk is to let more readers know

Comments 2:Field survey – sample points were chosen randomly and sampled regularly

Response 1:The sample points were randomly sampled according to the actual situation, because the dense vegetation and the intertidal zone were difficult to pass, we set random sample points evenly along the river.

Comments 3:Line 143-146 should be shifted to the discussion.

Response 3:Based on your suggestion, the section was moved to the discussion.

Comments 4:Fig. 2 Legend is very small and nearly unreadable. You can use the space on the bottom right and use only one legend, one scale and one north arrow

Response 4:We have tried to make the legend as large as possible in the picture, and you can see it clearly after magnification. Use the space on the bottom right and use only one legend, one scale and one north arrow would be incongruous.

Comments 5: Fig. 3 I miss significant level

Response 5: In Fig. 3 we analyzed the change trend of spartina area, but did not analyze the significance of differences.

Comments 6:Fig. 6 I miss descriptions of the axis and abbreviations of the habitat types

Response 6:We did not use abbreviations for habitat types, habitat type descriptions we present in the method.

Comments 7: I would start with change in spartina and why and then discuss methods

Response 7:Based on your suggestion, we start with change in spartina. The annual area of S. alterniflora decreased substantially in study area, which might be attributable to the foraging of elks and its trampling of spartina.

Comments 8:The discussion is mainly descriptive and there is no interpretation why breeding birds and/or migrants are affected. This may be discussed more deeply. 

Response 8:Based on your suggestion,  we discussion why breeding birds and/or migrants are affected. The breeding birds main comprises of songbirds (such as Sinosuthora webbiana) that breed in spartina and phragmites habitats, as well as a limited number of ground-nesting birds such as Vanellus cinereus. Most breeding songbirds are negatively affected by vegetation degradation. The habitat of ground-nesting birds increased, however their population remained stable. It may be that the number of elk was too large and frequent activity would increase the risk of tramping on bird nests on the ground. The primary avian migrants comprise wetland waterbirds, and the expansion of mudflat and water areas has resulted in increase of migrant populations.Migratory waterbirds usually gather in large groups around elk during the migration season, and migratory waterbirds may have strong adaptability to elk activities.

Comments 9: Conclusion is descriptive only. The results are described again but why does it happen and what are the consequences for the future?

Response 9: With the continuous degradation of vegetation and soil erosion, the number of waterbirds may decrease in the future.The excessive deposition of elk feces may potentially contribute to changes in soil and water quality to  change the habitat of waterbirds, thus necessitating further research for verification.

Reviewer 3 Report

Comments and Suggestions for Authors

Author Response

(The authors gave the same response as above.)

Reviewer 4 Report

Comments and Suggestions for Authors

I found the paper to be straightforward, and the topic has good management implications. I did not detect any problems with the methodology, analyses, or conclusions. I only had a handful of grammatical and stylistic changes.

Author Response

Thank you very much for taking the time to review this manuscript. According to your suggestions, we have made some grammatical and stylistic changes.